

**The importance of mineral determinations to PROFILE base**
**cation weathering release rates: A case study**
Sophie Casetou-Gustafson[1], Cecilia Akselsson[2], Stephen Hillier[1,3], Bengt A. Olsson[1],
[1]Department of Ecology, Swedish University of Agricultural Sciences, (SLU), P.O. Box 7044, SE-750 07
Uppsala, Sweden
[2]Department of Physical Geography and Ecosystem Science, Lund University, Sölvegatan 12, SE-223 62 Lund,
Sweden
[3]The James Hutton Institute, Craigiebuckler, Aberdeen AB15 8QH, United Kingdom
*Correspondence to*: Sophie Casetou-Gustafson (Sophie.Casetou@slu.se)





## 17 **Abstract**

This study explored the influence of uncertainty in quantitative mineralogy on PROFILE base cation (Ca, Mg, K,
Na) weathering rates obtained using normative mineralogy compared to those obtained using measured
mineralogy, which was taken as a reference. Weathering rates were determined for two sites, one in Northern
(Flakaliden) and one in Southern (Asa) Sweden. At each site, 3–4 soil profiles were analyzed at 10 cm depth
intervals. Normative quantitative mineralogy was calculated from geochemical data and qualitative mineral data
with the "Analysis to Mineralogy" program ('A2M') using two sets of qualitative mineralogical data inputs to
A2M: A site-specific mineralogy determined from X-ray powder diffraction (XRPD) analyses, and regional
mineralogy, representing the assumed mineral identity and compositions for larger geographical areas in Sweden.
For the site-specific mineral input the precise elemental compositions of minerals were determined by microprobe
analysis, whereas for the regional mineralogy the compositions were as assumed in previous studies. A2M does
not provide a unique mineralogical solution and one thousand random mineralogical solutions were calculated by
A2M for each soil unit in order to include the full space of quantitative mineralogies in model outcome, all equally
possible. A corresponding number of PROFILE runs were made to estimate weathering rates. The contribution of
individual minerals to the release of base cations was also quantified by using a version of PROFILE which outputs
this detail. A discrepancy between weathering rates calculated from XRPD data ($W_{XRPD}$) and weathering rates
based on A2M ($W_{A2M}$) was only considered significant if the former was outside the full range of the latter.
Arithmetic means of $W_{A2M}$ were generally in relatively close agreement with $W_{XRPD}$. The hypothesis that using
site-specific instead of regional mineralogy will improve the confidence in mineral data input to PROFILE was
supported for Flakaliden. However, at Asa, site-specific mineralogies reduced the discrepancy for Na between
$W_{A2M}$ and $W_{XRPD}$ but produced larger and significant discrepancies for K, Ca and Mg. For Ca and Mg the
differences between weathering rates based on different mineralogies could be explained by differences in the
content of some specific Ca- and Mg-bearing minerals, in particular amphibole, apatite, pyroxene and illite. It was
concluded that improving the precision in the content of those minerals would reduce weathering uncertainties.
High uncertainties in mineralogy, due for example to different A2M assumptions, had surprisingly low effect on
the weathering from Na- and K-bearing minerals. This can be explained by the fact that the weathering rate
constants for the minerals involved, e.g. K-feldspar and micas, are similar in PROFILE. Improving the description
of the dissolution rate kinetics of the plagioclase mineral group as well as major K-bearing minerals (K-feldspars
and micas) should be of particular importance to future weathering estimates.

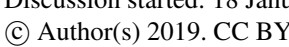



# Definitions and abbreviations

Mineralogy = the identity (specific mineral or mineral group) and stoichiometry (specific mineral chemical composition) of minerals that are present at a certain geographic unit, a particular site (*site-specific mineralogy*) or a larger geographic province (*regional mineralogy*)

Quantitative mineralogy= the quantitative information (wt.%) on the abundance of specific minerals in the soil.

**Abbreviations**:

$M_{XRPD}$ = quantitative mineralogy based on XRPD (amount) and electron microprobe analysis (composition)

$M_{A2M-reg}$ = quantitative mineralogy calculated with the A2M model and using regional mineralogy

$M_{A2M-site}$ = quantitative mineralogy calculated with the A2M model and using site-specific mineralogy

$W_{XRPD}$ = weathering rate based on quantitative mineralogy determined by XRPD and electron microprobe analysis

$W_{A2M}$ = weathering rate based on quantitative mineralogy determined by the A2M model (unspecific mineralogy)

$W_{A2M-reg}$ = weathering rate based on quantitative mineralogy determined by the A2M model, and assuming regional mineralogy.

$W_{A2M-site}$ = weathering rate based on quantitative mineralogy determined by the A2M model and assuming site-specific mineralogy.



## 1. Introduction

The dissolution of minerals during weathering represents, together with deposition, the most important long-term supply of base cations for plant growth as well as acting as a buffer against soil and water acidification. Quantifying weathering rates is therefore of key importance to guide modern forestry demands on biomass removal by helping to identify threshold levels of sustainable base cation removal from soils and waters. With the introduction of the harvest of forest biomass for energy production that includes whole tree harvest and stump extraction, about 2–3 times more nutrients are exported from the forest compared to stem-only harvest. As a result, issues of acidification and base cation supply are exacerbated and the sustainability of this practice is questioned (Röser, 2008; de Jong et al. 2017). Regional nutrient balance calculations for Sweden have indicated that net losses of base cations from forest soils can occur in stem-only harvest scenarios, and this trend was substantially exacerbated and became more frequent in whole-tree harvesting scenarios, largely due to low weathering rates (Sverdrup and Rosén, 1998; Akselsson et al., 2007a,b). The same effect occurred both under current and projected future climate conditions (Akselsson et al., 2016).

The weathering rates included in these nutrient balance calculations are in most cases based on the PROFILE model. This model is the most used mechanistic tool to calculate steady state chemical weathering at the interface of soil minerals and their surrounding liquid solution (Sverdrup, 1996), and has been widely applied in Europe and the US during the last several decades or more of weathering research (Olsson et al., 1993; Langan et al., 1995; Kolka et al., 1996; Starr et al., 1998; Sverdrup and Rosén, 1998; Koseva et al., 2010; Whitfield et al., 2006; Akselsson et al., 2007a; Stendahl et al., 2013). In a few cases nutrient balance calculations have also been based on the depletion method (Olsson et al., 1993).

Reliable weathering rate estimates are crucial for the accuracy of future nutrient budget calculations (Futter et al., 2012). Regarding the accuracy of the PROFILE model, the importance of high accuracy in physical input parameters for the modelled weathering rate outputs has been highlighted by Hodson et al. (1996) and Jönsson et al. (1995). Among these parameters Hodson et al. (1996) noted that the weathering response of the entire soil profile depends critically on its mineralogy. However, little attention has been given to the influence of modelled versus directly measured mineralogical input data on calculated base cation release rates.

The most widely used method for direct quantitative mineralogical analysis of soil samples is X-ray powder diffraction, and the accuracy that can be achieved has been demonstrated in round robin tests most notably the Reynolds Cup (McCarty, 2002; Kleeberg, 2005; Omotoso et al., 2006, Raven and Self, 2017). Casetou-Gustafson et al. (2018) made some independent assessment of the accuracy of their own XRPD data by geochemical cross validation (i.e. the mineral budgeting approach of Andrist-Rangel et al., 2006). Nonetheless, we should stress that like all analytical methods the determined weight fractions of minerals identified in a soil sample by XRPD will have an associated uncertainty, and additionally minerals present in minor amounts, nominally < 1% by weight, may fall below the lower limit of detection of the XPRD method.

Due mainly to the relative ease of measurement and consequent ready availability of total element geochemical data on soils, indirect methods of determining quantitative soil mineralogy, such as so called 'normative' geochemical calculations have been widely used to generate mineralogical data for use in the PROFILE model. One such method is the normative "Analysis to Mineralogy" (A2M) program (Posch and Kurz, 2007) that has



commonly been used in PROFILE applications (Stendahl et al. 2013; Zanchi et al. 2014, this issue; Yu et al. 2016;
2018; Kronnäs et al., 2019). Based on a quantitative geochemical analysis of a soil sample, typically expressed in
weight percent oxides, and also on some assessment of the available minerals in the soil sample (minerals present)
and their stoichiometry (chemical compositions), A2M calculates all possible mineralogical compositions. The
A2M output for a given soil sample input has multiple solutions and can be described as a multidimensional
mineralogical solution space. This necessitates a choice when using A2M output in applications such as weathering
rate studies, the convention for which has been to use the geometric mean mineralogical compositions e.g. Stendahl
et al. 2013. Casetou-Gustafson et al. (2018) compared the output of A2M with directly determined XRPD
mineralogies at two sites, applying A2M in two different ways. In the first case the information on available
minerals in the model input was obtained from direct XRPD mineral identifications and information on mineral
stoichiometry from direct microprobe analysis of the minerals at the specific site (hereafter denoted "site-
specific"). In the second case the mineral stoichiometry and mineral identity were both assumed based on an expert
assessment of the probable mineralogy at the regional scale as given by Warfvinge and Sverdrup (1995), hereafter
denoted "regional". Casetou-Gustafson et al. (2018) concluded that using A2M in combination with regional input
data yielded results with large deviations from directly (XRPD) measured quantitative mineralogy, particularly for
two of the major minerals, K-feldspar and dioctahedral mica. When site-specific mineralogical input data was
used, measured and modeled quantitative mineralogy showed a better correspondence for most minerals. For a
specific mineral and a specific site, however, the bias in determination of quantitative mineralogy might be
significant depending on the accuracy of input data to A2M, i.e. total geochemistry and/or mineral stoichiometry
(Casetou-Gustafson et al., 2018). Errors like these in mineralogical input data might be assumed to affect the
calculated weathering for different base cations significantly.
In the present study, we used the different mineralogical data from Casetou-Gustafson et al. (2018) to model
weathering rates of soils with the PROFILE model. Rates calculated based on measured mineral abundances using
quantitative XRPD in combination with measured elemental compositions are taken as 'reference' weathering
rates to which other rates are compared. Samples for this study were collected from podzolised till soils from 8
soil profiles at two forest sites in northern and southern Sweden, respectively.
The primary objective of this study was to describe and quantify the effect of differences in mineralogy input on
PROFILE weathering rates, leaving all other input parameters of the PROFILE model constant to isolate the effects
of mineral stoichiometry and abundance. A first specific aim was to determine the uncertainties in weathering rates
caused by uncertainties in normative quantitative mineralogy. This was approached by comparing PROFILE runs
using modeled mineralogies based on the presence of minerals of a specific site or a larger geographic region (i.e.
site-specific and regional mineralogy) with PROFILE runs using the directly measured mineralogy. The latter was
assumed to represent the 'true' mineralogy at each site. This comparison of PROFILE weathering rates, based on
XRPD versus A2M mineralogy, was done using 1000 random solutions per sample from the entire
multidimensional A2M solution space. In the following, weathering rates calculated by PROFILE based on XRPD
and A2M mineralogies are denoted $W_{XRPD}$ and $W_{A2M}$, respectively.
A second specific aim was to investigate how the over- or underestimation of $W_{A2M}$ in relation to $W_{XRPD}$ mirrors
the over- or underestimation of mineral contents estimated with A2M.

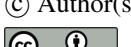



The following hypotheses were made:
(1) PROFILE weathering rates obtained with site-specific mineralogical data are more similar to the reference
weathering rates than PROFILE runs obtained with regional mineralogical input data
(2) Over- and underestimations of weathering rates of different base cations by the PROFILE model can be
explained by over- and underestimations of mineral contents of a few specific minerals.

**150    2. Materials and methods**

**151    2.1 Study sites**

Two experimental forest sites, Asa in southern, and Flakaliden in northern Sweden, were used for the study (Table
1). Both sites have Norway spruce (*Picea abies* (L.) Karst) stands of uniform age, but differ in climate. Flakaliden
is located in the boreal zone with long cold winters, whereas Asa is located in the hemiboreal zone. The soils have
similar texture (Sandy loamy till), soil types (Spodosols) and moisture conditions. According to the geographical
distribution of mineralogy types in Sweden the sites belong to different regions (Warfvinge and Sverdrup, 1995).
The experiments, which started in 1986, aimed at investigating the effects of optimized water and nutrient supply
on tree growth and carbon cycling in Norway spruce forests (Linder 1995, Albaugh et al. 2009). The sites are
incorporated in the Swedish Infrastructure for Ecosystem Science (SITES).

**160    2.2 Soil sampling and stoniness determination**

Soil sampling was performed in October 2013 and March 2014 in the border zone of four plots each of the sites.
Plots selected for sampling were untreated control plots (K1 and K4 at Asa, 10B and 14B at Flakaliden) and
fertilized 'F' plots (F3, F4 at Asa, 15A, 11B at Flakaliden). A rotary drill was used in order to extract one intact
soil core per plot (17 cm inner diameter) expect for plot K4, F3 and F4 at the Asa site. A 1 x 1m soil pit was
excavated at each of the three latter plots due to inaccessible terrain for forest machinery. The maximum mineral
soil depth varied between 70–90 cm in Flakaliden and 90–100 cm in Asa.

The volume of stones and boulders was determined with the penetration method by Viro (1952), and by applying
penetration data to the functions by Stendahl et al. (2009). A metal rod was penetrated at 16 points per plot into
the soil until the underground was not possible to penetrate any further, or to the depth 30 cm. There was a higher
average stoniness at Flakaliden than Asa (39 vol-% compared to 29 vol-% in Asa) that could partially explain the
lower maximum sampling depth at Flakaliden.

**173    2.3 Sample preparation**

Soils samples for chemical analyses were taken at 10 cm depth intervals in the mineral soil. Prior to analysis all
soil samples were dried at 30–40 °C and sieved at 2 mm mesh. Soil chemical analyses were performed on the fine
earth fraction (< 2mm).





### 2.4 Analysis of geochemistry, total carbon and soil texture


Total carbon was determined using a LECO elemental analyzer according to ISO 10694. Analysis of total
geochemical composition, conducted by ALS Scandinavia AB, was made by inductively coupled plasma
spectrometry (ICP-MS). Prior to analyses, the samples were ignited at 1000° C to oxidize organic matter and
grinded with an agate mortar. Particle size distribution was analyzed by wet sieving and sedimentation (Pipette
method) in accordance with ISO 11277. More details about the analytical procedure was given by Casetou-
Gustafson et al. (2018).

### 2.5 Determination of quantitative mineralogy


A detailed description of methods used to quantify mineralogy of the samples were given by Casetou et al.
(2018), and are described in brief below.

### 2.5.1 Measured mineralogy


Quantitative soil mineralogy was determined with the X-ray powder diffraction technique, XRPD ($M_{XRPD}$) (Hillier
1999, 2003). Preparation of samples for determination of XRPD patterns was made from spraying and drying
slurries of micronized soil samples (<2 mm) in ethanol. Quantitative mineralogical analysis of the diffraction data
was performed using a full pattern fitting approach (Omotoso et al., 2006). In the fitting process, the measured
diffraction pattern is modelled as a weighted sum of previously recorded and carefully verified standard reference
patterns of the prior identified mineral components. The chemical composition of the various minerals present in
the soils was determined by electron microprobe analysis (EMPA).

### 2.5.2 Calculated mineralogy


The A2M program (Posch and Kurz, 2007) was used to calculate quantitative mineralogical composition ($M_{A2M}$)
from geochemical data. Based on a set of pre-determined data on mineral identity and stoichiometry, the model
outcome is a range of equally possible mineralogical compositions. The multidimensional structure of this
normative mineralogy model is a consequence of the number of minerals being larger than the number of analysed
elements, where a specific element can often be contained in several different minerals. A system of linear
equations is used to construct an M-N dimensional solution space (Dimension M= Number of minerals, Dimension
N=number of oxides). In this study we used one thousand solutions to cover the range of possible quantitative
mineralogies that may occur at a specific site.

A2M was used to calculate 1000 quantitative mineralogies each for two different sets of mineral identity and
element stoichiometry, $M_{A2M-reg}$ (regional) and $M_{A2M-site}$ (site-specific). Regional mineralogy refers to the mineral
identity and stoichiometry for the four major mineralogical provinces in Sweden suggested by Warfvinge and
Sverdrup (1995), of which Asa and Flakaliden belong to different regions. Site-specific mineralogy refers to the
measured mineral identity and stoichiometry determined by the XRPD and electron microprobe analyses of the
two sites (Casetou-Gustafson et al., 2018).



### 2.6. Estimation of weathering rates with PROFILE

#### 2.6.1 PROFILE model description

The biogeochemical PROFILE model can be used to study the steady state weathering (i.e. stoichiometric mineral dissolution) of soil profiles, as weathering is known to be primarily determined by the physical soil properties at the interface of wetted mineral surfaces and the soil solution. PROFILE is a multilayer model, thus, for each soil layer, parameters are specified based on field measurements and estimation methods (Warfvinge and Sverdrup, 1995). Furthermore, isotropic, well mixed soil solution conditions are assumed to prevail in each layer as well as surface limited dissolution in line with early views by Aagard and Helgeson (1982) and Cou and Wollast (1985) (Sverdrup, 1996). Based on these major assumptions, PROFILE calculates chemical weathering rates from a series of kinetic reactions that are described by laboratory determined dissolution rate coefficients and soil solution equilibria (i.e. transition state theory) (Sverdrup and Warfvinge, 1993). The PROFILE version (September 2018) that was used in this study is coded to produce information on the weathering contribution of specific minerals, which allowed us to test our second hypothesis. This version is based on the weathering rates of 15 minerals. Of these, apatite, pyroxene, illite, dolomite and calcite were not found at the two study sites according to XRPD data (Table S1).

#### 2.6.2 PROFILE parameter estimation

The only parameter that was changed between different PROFILE runs was the quantitative mineralogy for each soil layer, as described above. Hence, PROFILE estimated weathering rates (W) based on measured mineralogy ($W_{XRPD}$), and the two versions of A2M calculated mineralogy, regional ($W_{A2M-reg}$), and site-specific ($W_{A2M-site}$). In the regional mineralogy, plagioclase is assumed to occur as pure anorthite and albite for simplification as has been used in previous studies (Stendahl et al., 2013; Zanchi et al., 2014). This simplification was done in order to avoid having a number of minerals containing different amount of Ca and Na, as a result of plagioclase forming a continuous solid solution series, since it would not affect the weathering rates.

The physical soil layer specific parameters, that were kept constant between different profile runs, were exposed mineral surface area, stoniness, soil bulk density and soil moisture (Table 2). Exposed mineral surface area was estimated from soil bulk density and texture analyses in combination with an algorithm specified in Warfvinge and Sverdrup (1995). The volumetric field soil water content in Flakaliden and Asa was estimated to be 0.25 $m^3$ $m^{-3}$ according to the moisture classification scheme described in Warfvinge and Sverdrup (1995). It was used to describe the volumetric water content for each soil pit.

Another group of parameters kept constant was chemical soil layer specific parameters. Aluminum solubility coefficient needed for solution equilibrium reactions, defined as $\log\{Al^{3+}\}+3pH$, was estimated applying a function developed from previously published data (Simonsson and Berggren, 1998) on our own total carbon and oxalate extractable aluminum measurements. The function is based on the finding that the Al solubility in the upper B-horizon of Podzols is closely related to the molar ratio of aluminum to carbon in pyrophosphate extracts, and that below the threshold value of 0.1, Al solubility increases with the $Al_p/C_p$ ratio (Simonsson and Berggren, 1998). Thus, a function was developed for application to our own measurements of $Al_{ox}$ and $C_{tot}$ based on the assumption that it is possible to use the $Al_{ox}/C_{tot}$ ratio instead of the $Al_p/C_p$ ratio. Data on soil solution DOC were





available from lysimeters installed at 50 cm depth for plot K4 and K1 in Asa and 10B and 14B in Flakaliden, and
these values were also applied to soil depths below 50 cm (H. Grip, unpublished data). The E-horizon (0 –10 cm
at Flakaliden) and A-horizon (0 –10 cm at Asa) were characterized by higher DOC values based on previous
findings (Fröberg et al., 2013) and the classification scheme of DOC in Warfvinge and Sverdrup (1995). Partial
$CO_2$ pressure values in the soil were taken from the default estimate of Warfvinge and Sverdrup (1995).

Other site-specific parameters that were kept constant between PROFILE runs were evapotranspiration,
temperature, atmospheric deposition, precipitation, runoff and nutrient uptake. An estimate of the average
evaporation per site was derived from annual averages of precipitation and runoff data using a general water
balance equation. Deposition data from two sites of the Swedish ICP Integrated Monitoring catchments, Aneboda
(for Asa) and Gammtratten (for Flakaliden) (Löfgren et al., 2011) were used to calculate the total deposition. The
canopy budget method of Staelens et al. (2008) was applied as in Zetterberg et al. (2014) for $Ca^{2+}$, $Mg^{2+}$, $K^+$, $Na^+$.
The canopy budget model is commonly used for elements that are prone to canopy leaching ($Ca^{2+}$, $Mg^{2+}$, $K^+$, $Na^+$,
$SO_4^{2-}$) or canopy uptake ($NH_4^+$, $NO_3^-$) and calculates the total deposition (TD) as the sum of dry deposition (DD)
and wet deposition (WD). Wet deposition was estimated based on the contribution of dry deposition to bulk
deposition, both for base cations and anions, using dry deposition factors from Karlsson et al. (2012). Base cation
and nitrogen accumulation rate in above-ground tree biomass (i.e. bark, stemwood, living and dead branches,
needles) was estimated as the average accumulation rate over a 100 years rotation length in Flakaliden compared
to a 73 years rotation length in Asa. These calculations were based on Heureka simulations using the StandWise
application (Wikström et al., 2011) for biomass estimates in combination with measured nutrient concentrations
in above- ground biomass (S. Linder unpubl. data).

**271    2.7 A definition of significant discrepancies between $W_{A2M}$ and $W_{XRPD}$**

A consequence of the mathematical structure of the A2M program is that the space of possible quantitative
mineralogies in the end produces an uncertainty range of weathering estimates, but in a different sense than the
uncertainty caused by e.g. uncertainties in chemical analyses, because all mineralogies produced within this range
are equally likely. Thus, here we define a significant discrepancy between $W_{XRPD}$ and $W_{A2M}$ to occur when the
former is outside the range of the latter, as illustrated in Fig. 1a. The opposite case is a non-significant discrepancy,
when the weathering rates based on XPRD are contained in the weathering range based on A2M (Figure 1b).

The uncertainty range of $W_{A2M}$ can potentially be reduced by reducing uncertainties in analyses of soil
geochemistry but most particularly by definitions of available minerals and their stoichiometry. Furthermore, some
discrepancies between $W_{XRPD}$ and $W_{A2M}$ might also arise due to limitations of the XRPD method, particularly
when minerals occur near or below the detection limit.

**283    2.8 Statistical analyses**

In order to quantify the effect of mineralogy on PROFILE weathering rates two statistical measures were used to
describe the discrepancies between $W_{XRPD}$ and $W_{A2M}$. Firstly, root mean square errors (RMSE) of the differences
between $W_{XRPD}$ and the arithmetic mean of weathering rates based on regional and site-specific mineralogy, i.e.,
$W_{A2M-reg}$ and $W_{A2M-site}$, were calculated:





$RMSE = \sqrt{\frac{1}{n}\sum_{i=1}^{n}(WXRPD) - WA2M)^2_i}$          Eq. (1)
RMSE's were calculated individually for each element, soil layer and soil profile for two data sets. An RMSE
expressing the error of the aggregated, total weathering rates in the 0–50 cm soil horizon was calculated to test our
first hypothesis (RMSE of total weathering). In addition, an RMSE expressing the errors originating from
discrepancies between $W_{XRPD}$ and $W_{A2M}$ for individual minerals was also calculated (RMSE of weathering by
mineral). In the latter case, sums of RMSE's by mineral were calculated for each element and soil profile in analogy
with the summing up of weathering rates for the whole 0–50 cm soil profile.

Secondly, relative discrepancies (i.e. average percentage of over- or underestimation of $W_{A2M}$ compared to $W_{XRPD}$)
were calculated as the absolute discrepancy divided by the measured value.

Relative error $= (\frac{(WA2M)_i - WXRPD)_i}{WXRPDi}) * 100$          Eq. (2)

Relative errors were calculated for each site by comparing the average sum of $W_{A2M}$ in the upper mineral soil (0–
50 cm) with the sum of $W_{XRPD}$ in the upper mineral soil.
Statistical plotting of results was performed using R (version 3.3.0) (R Core Team, 2016) and Excel 2016.

**304**     **3. Results**

**305**     **3.1 Weathering rates based on XRPD mineralogy**

Weathering estimates with PROFILE are hereafter presented as the sum of weathering rates in the 0–50 cm soil
horizon, since this soil depth is commonly used in weathering rate studies. Information on individual, and deeper
soil layers (50-100 cm) is given in Table S2.

Weathering rates of the base cations based on quantitative XRPD mineralogy ($W_{XRPD}$), the reference weathering
rates, were ranked in the same order at both sites, i.e., Na>Ca>K>Mg (Table S2). On average, weathering rates of
Na, Ca, K and Mg at Asa were 17.7, 8.4, 5.6 and 3.6 $mmol_c$ $m^{-2}$ $yr^{-1}$, respectively. Corresponding figures for
Flakaliden were of similar magnitude, i.e., 14.8, 9.8, 5.7 and 5.6 $mmol_c$ $m^{-2}$ $yr^{-1}$. The variation in weathering rates
between soil profiles was smaller at Asa than at Flakaliden, as the standard deviation in relation to the means for
different elements ranged between 0.2-2.3 at Asa, and 2.0 –5.7 at Flakaliden (Table S2).

**317**     **3.2 Comparison between weathering rates based on XRPD and A2M mineralogy**

At Flakaliden, $W_{A2M-site}$ was generally in closer agreement with $W_{XRPD}$ than $W_{A2M-reg}$ (Fig. 2b), in line with the first
hypothesis. The discrepancies between $W_{XRPD}$ and $W_{A2M}$ were small and non-significant for Mg regardless of the
mineralogy input used in A2M, although the estimated discrepancies were reduced when site-specific mineralogy
was used. The use of regional mineralogy in A2M underestimated K release rates compared to $W_{XRPD}$, and the
discrepancy was significant. Using site-specific mineralogy resulted in smaller and non-significant discrepancy
for K release rates. A similar response to different mineralogies was revealed for Ca, although the result varied
more among soil profiles. In contrast to K and Ca, the release of Na was overestimated by both $W_{A2M-site}$ and $W_{A2M-}$



$_{reg}$ compared to $W_{XRPD}$. The discrepancies were significant regardless of the mineralogy input used in A2M,
although using site-specific mineralogy slightly reduced the discrepancy. The generally closer agreement between
$W_{A2M\text{-}site}$ and $W_{XRPD}$ than $W_{A2M\text{-}reg}$ at Flakaliden was also indicated by the lower RMSEs of total weathering for all
base cations when site-specific mineralogy was used (Fig. 3a). Relative RMSE were below 20 % for $W_{A2M\text{-}reg}$, but
below 10 % for $W_{A2M\text{-}site}$. However, RMSE for Na was only slightly smaller for $W_{A2M\text{-}site}$ than $W_{A2M\text{-}reg}$ (16 % for
$W_{A2M\text{-}site}$).

PROFILE weathering rates for Asa revealed a different pattern compared to Flakaliden, and the results for Ca, Mg
and K were contradictory to hypothesis one. $W_{A2M\text{-}reg}$ was in close agreement with $W_{XRPD}$ for K, Ca and Mg, and
the small discrepancies were non-significant (Fig. 2a). However, $W_{A2M\text{-}reg}$ for Na was consistently overestimated
compared to $W_{XRPD}$ and the discrepancies were significant. Using site-specific mineralogy improved the fit
between $W_{XRPD}$ and $W_{A2M}$ for Na, but had rather the opposite effect on the other base cations at this site. For K,
Ca and Mg, $W_{A2M\text{-}site}$ overestimated weathering rates, and resulted in significant discrepancies, and larger RMSE,
whereas the discrepancies for Na were reduced and non-significant (Fig. 3a). At Asa, the highest relative RMSEs
of total weathering occurred for Ca and Mg with $W_{A2M\text{-}site}$ (> 30 %) (Fig. 3a). Large standard deviations were due
to a single soil profile, F4. The better consistency with $W_{A2M\text{-}reg}$ was indicated by RMSE below 10 % for Ca and
Mg, and that RMSE for Mg was half of the error with $W_{A2M\text{-}site}$. Only for Na, RMSE was lower for $W_{A2M\text{-}site}$ than
with $W_{A2M\text{-}reg}$.

A complementary illustration of the relationships between weathering rates based on XRPD and A2M is shown in
Fig. 4 and provided as Tables S3 and S4, which includes all data from individual soil layers 0–50 cm. A general
picture is that $W_{A2M\text{-}site}$ was less dispersed along the 1:1-line than $W_{A2M\text{-}reg}$, in particular for Flakaliden. On the
other hand, for weathering rates in the lower range (< 5 $mmol_c$ m$^{-2}$ yr$^{-1}$) site-specific mineralogy tended to generate
both over- and underestimated weathering rates. In most soil profiles, deviations from the 1:1-line were frequent
in soil layers below 20 cm. For Na under- and overestimations occurred in the whole range of weathering
estimates,

**351 3.3 Mineral-specific contribution to weathering rates**

In spite of its intermediate dissolution rate plagioclase was, due to its abundance, the most important Na-bearing
mineral determined in this study (Table 3 and Fig. 5). Plagioclase is a variable group of minerals with different
stoichiometric proportions of Ca and Na, from the purely sodic albite on the one hand to the purely calcic anorthite
on the other hand (Table S5) as well as with intermediate compositions (Table S6). For simplicity, they will be
referred to in this study as sodic and calcic plagioclase. Based on the same quantitative mineralogy (i.e. same
elemental compositions and identity of minerals), $W_{XRPD}$ and $W_{A2M\text{-}site}$ gave strong weight to both calcic and sodic
plagioclase in estimating Na release rates, but $W_{A2M\text{-}site}$ gave stronger weight to calcic versus sodic plagioclase at
Asa, and vice-versa at Flakaliden (Fig. 5). In spite of these differences, the resultant release rates of Na according
to $W_{A2M\text{-}site}$ and $W_{XRPD}$ were rather similar (Fig. 5).

Total Na release rates of $W_{A2M\text{-}reg}$ compared to $W_{XRPD}$ were moderately overestimated. The relative RMSE of
weathering by specific Na-containing minerals were of more similar magnitude for Na at Flakaliden compared to





Asa (Fig. 3b). However, the standard deviations of RMSE were relatively large at Flakaliden, due to large RMSE
for albite in one specific soil profile (11B) (Table S7). Contrary to relative RMSE of total weathering, the relative
RMSE of weathering by specific minerals was lower for Na at Asa with regional than site-specific mineralogy.
According to $W_{XRPD}$, calcic plagioclase weathering was the most important source to Ca release at Flakaliden, and
the second most important source at Asa after epidote (Fig. 5). As for Na, $W_{A2M-site}$ gave stronger weight to calcic
plagioclase than $W_{XRPD}$ at Asa. It was the other way around for $W_{A2M-site}$ at Flakaliden and the regional mineralogy
(i.e. $W_{XRPD}$ stronger weight to calcic plagioclase than $W_{A2M-site}$). Another important Ca source in weathering
estimates based on A2M was apatite. This mineral was not detected in the XRPD analyses, but was included in
both A2M mineralogies as a necessary means to allocate measured total phosphorus content to a specific mineral
(Casetou-Gustafson et al. 2018).
Similar to Na, relative RMSE of weathering by Ca-containing minerals were several magnitudes larger than RMSE
of the total weathering of Ca. In other words, although an overall similar weathering rates might be generated by
the PROFILE model based on different quantitative mineralogies, the underlying contributions from different
minerals can be markedly different. At Flakaliden, the mean relative RMSE by specific minerals were larger for
regional than site-specific mineralogy at Flakaliden (Fig. 3b). However, the difference was not significant since
the standard deviations were high, probably due to larger RMSE for Ca-bearing minerals in soil profile 11B (Table
S7).
A general picture of the mineral contribution to Mg release is that $W_{XRPD}$ placed most weight to amphibole whereas
in $W_{A2M}$ Mg release was more equally distributed among other minerals, notably hydrobiotite, trioctahedral mica
and vermiculite. At Asa, and to an even larger extent at Flakaliden, Mg release by A2M mineralogies was
determined by a higher contribution of minerals with high dissolution rates (Fig. 5 and Table 3) (i.e. In $W_{A2M-site}$,
hydrobiotite and trioctahedral mica; In $W_{A2M-reg}$, muscovite and vermiculite at Asa .and biotite and illite at
Flakaliden). At Asa, less weight was given to amphibole by $W_{A2M-site}$ compared to $W_{XRPD}$. At Flakaliden, the $W_{A2M-site}$ was close $W_{XRPD}$ in spite of the very different allocations of weathering rates to different minerals. The
underestimation of Mg release by $W_{A2M-reg}$ was largely explained by the lower weight given to amphibole in both
A2M scenarios (Fig. 5). However, A2M gave larger weight to other minerals. The sums of RMSEs of weathering
from specific Mg-bearing minerals were much larger for regional than site-specific mineralogy at Flakaliden and
reached a maximum value of 156 %. A contributing factor were generally larger RMSE for the mineral
contribution of amphibole to Mg weathering and the fact that pyroxene contributed to the RMSEs of the total
weathering of Mg. Furthermore, a large standard deviation for the sum of RMSE of specific minerals (Fig. 3b)
was caused by soil profile 11B where more weight was placed on amphibole and biotite in contributing to Mg
weathering (Table S7). The two A2M mineralogies resulted in the same RMSEs for Mg-bearing minerals at Asa
(Fig. 3b).
Potassium release rates were largely dominated by K-Feldspar weathering in both $W_{XRPD}$ and $W_{A2M-site}$. However,
K release by $W_{A2M-reg}$ (Fig. 5) were largely determined by micas at both sites. Together with Mg, these elements
had also the lowest weathering rates, indicating that differences between $W_{A2M-reg}$ and $W_{XRPD}$ in relative terms were





not correlated with the magnitude of weathering. Unlike the other base cations, relative RMSE of K-bearing
minerals were lower at both sites when site-specific mineralogy was used instead of regional (Fig. 3b), and the
mineral specific RMSEs were also of similar magnitude as the RMSE of the total weathering (Fig.3a). $W_{A2M\text{-site}}$ of
K (Fig. 3b), were not several magnitudes larger than RMSE of the total weathering (Fig. 3a). The largest relative
RMSEs of K-containing minerals were reached by $W_{A2M\text{-reg}}$ at Flakaliden in soil profile 11B, indicated by the high
standard deviation.
**4. Discussion**
**4.1 General range of weathering rates in relation to expectations from other sensitivity studies, and the**
**range of discrepancies between $W_{XRPD}$ and WA2M**
To our knowledge, the present study is the first to have examined the sensitivity of the PROFILE model on real
case study differences of directly measured mineralogy versus indirectly determined normative mineralogy.
However, a few systematic studies have been made previously to test the influence of mineralogy inputs, amongst
other input parameters, to PROFILE weathering rates. Jönsson et al. (1995) concluded that uncertainty in
quantitative mineralogy could account for a variation from the best weathering estimate of about 20 %, and that
variations in soil physical and chemical parameters could be more important. The sensitivity analysis of Jönsson
et al. (1995) was made by a Monte Carlo simulation where mineralogical inputs were varied by ± 20 % of abundant
minerals, and up to ± 100 % of minor minerals. Shortly after, Hodson et al. (1996) examined the sensitivity of the
PROFILE model with respect to the sensitivity of weathering of specific minerals and concluded that large
uncertainties in particular in soil mineralogy, moisture, bulk density, temperature and surface area determinations
will have a larger effect on weathering rates than was reported by Jönsson et al. (1995).
Compared with the sensitivity analyses by Jönsson et al. (1995), the range of uncertainty in dominating mineral
inputs used in the present study was of similar order of magnitude. For this study we used the XRPD measured
($M_{XRPD}$) and A2M estimated mineralogies ($M_{A2M}$) determined by Casetou-Gustafson et al. (2018). For example,
they concluded that $M_{A2M\text{-reg}}$ produced a low relative RMSE of total plagioclase (7 – 11 %) but higher relative
RMSE for less abundant minerals, such as dioctahedral mica (90 – 106 %). They also showed that when regional
mineral identity and assumed stoichiometry was replaced by site-specific mineralogy ($M_{A2M\text{-site}}$), the bias in
quantitative mineralogy was reduced.
Thus, given this bias in quantitative mineralogy input to PROFILE, discrepancies of $W_{A2M}$ from $W_{XRPD}$ at our
study sites should have been on the order of 20 % or less, and site-specific mineralogy inputs should produce
weathering rates with lesser discrepancies than regional mineralogy. The result of this study was in agreement
with this expectation for all elements at Flakaliden but only for Na at Asa. The different quantitative mineralogies
resulted in discrepancies between $W_{A2M}$ and $W_{XRPD}$ that differed with site (Fig. 3a, 5).
**4.2 Is $W_{A2M\text{-site}}$ more consistent than $W_{A2M\text{-reg}}$?**
Our first hypothesis, that using site-specific mineralogy in the PROFILE model compared to regional mineralogy,
should result in weathering rates closer to XRPD-based mineralogy, and thus be more consistent, was generally
supported for Flakaliden, but only for Na at Asa. This result was revealed from both the occurrence of significant
discrepancies as well as the RMSE of the total weathering rates. Thus, the results did not support our first



hypothesis in a consistent way. The possible reasons for this outcome are discussed below, based on the analysis
of how different minerals contributed to the overall weathering rates.

**4.3 How are discrepancies between $W_{A2M}$ and $W_{XRPD}$ correlated to bias in determinations of quantitative**
**443**
**444 mineralogy**

The version of the PROFILE model used in this study allowed a close examination of the per element weathering
rate contributions obtained from different minerals that provide some insight into the causes to the total $W_{A2M}$
discrepancies.

**448 4.3.1 Sodium release rates**

A biased determination of mineralogy may not necessarily result in a corresponding bias of PROFILE weathering
estimates if the discrepancies are cancelling each other out, and if dissolution rates of the different minerals are
rather similar. This was probably the case for Na. At both study sites and for both $W_{XRPD}$ and $W_{A2M}$, Na release
rates were largest for plagioclase minerals. The Na release from $W_{A2M-site}$ and $W_{A2M-reg}$ were close to $W_{XRPD}$ at both
study sites (i.e. all weathering rates were in the range of 17-19 mmol$_c$ m$^{-2}$ yr$^{-1}$) in spite of that $W_{A2M-site}$ placed
more weight to calcic plagioclase and $W_{A2M-reg}$ more weight to albitic plagioclase (Fig.5). Contrary to our second
hypothesis, the relatively high precision in total release rates (i.e.<10%; Fig. 3a) of Na was not correlated to the
actual low precision in mineral contribution to the total Na release rates (i.e. >30 %; Fig. 3b). The latter can be
explained by the fact that in PROFILE all types of plagioclase have the same dissolution rate coefficients (Table
3). Due to this, and in combination with the fact that plagioclase type minerals are a major source for Na, the
mineralogical uncertainty in estimating Na release rates with PROFILE was relatively low in this study (i.e. <20
%). In context, however, we note that it is generally accepted that under natural conditions different plagioclase
minerals weather at different rates, (Allen and Hajek, 1989, Blum and Stillings, 1995).

**462 4.3.2 Calcium release rates**

According to $W_{XRPD}$ and $W_{A2M}$, a key mineral for Ca release rates was calcic plagioclase at Flakaliden and epidote
at Asa. In line with our second hypothesis, the overestimation of calcic plagioclase in $M_{A2M-site}$ at Asa at the expense
of epidote and amphibole (Casetou-Gustafson et al., 2018) was directly reflected in the significant discrepancy
and overestimated weathering rates of Ca by $W_{A2M-site}$ compared to $W_{XRPD}$ (Fig. 5, and 1a). This discrepancy was
due to differences between $W_{A2M-site}$ and $W_{XRPD}$ in the mineral stoichiometry of calcic plagioclases, and not in
geochemistry, as the same geochemical analyses were also used for $W_{A2M-reg}$.

At Flakaliden, A2M based on site-specific mineralogy overestimated epidote at the expense of amphibole
(Casetou-Gustafson et al., 2018), leading to an underestimation of Ca weathering rates from amphibole compared
to epidote (Fig. 5). On the other hand, at Asa, it was the regional mineralogy input to A2M that resulted in
overestimated amounts of epidote at the expense of dioctahedral vermiculite and amphibole, and this bias was
directly reflected in the underestimated release of Ca from amphibole in $W_{A2M-reg}$. Conversely, the relatively small
and non-significant discrepancies of Ca release by $W_{A2M-site}$ at Flakaliden and by $W_{A2M-reg}$ at Asa did not depend
on a high precision in estimating the contribution from different minerals, since the precision was actually low. In
these cases, the good fits seem to be simply coincidental. Owing to differences in dissolution rates, Ca-bearing
minerals tend to compensate each other in terms of the total weathering rate that is calculated. This compensatory





effect is perhaps the reason why by coincidence, both $W_{A2M-reg}$ and $W_{A2M-site}$ discrepancies for Ca diverge in
different directions at Asa compared to Flakaliden.

Another source of uncertainty associated with the release of Ca is the role of minerals with high dissolution rates
that occur in low abundance, for example apatite and pyroxene. Apatite was included in $M_{A2M}$, but if present in the
soils studied was below the detection limit of 1 wt.% in the XRPD analyses (Casetou-Gustafson et al., 2018).
Additionally,  the assumption made in the A2M calculations that all P determined in the geochemical analyses is
allocated to apatite will likely overestimate the abundance of this mineral since soil P can also occur bound to Fe
and Al oxides and soil organic matter in acidic mineral soils (Weil and Brady, 2016). The high occurrence of
paracrystalline Fe-oxyhydroxide and Al-containing allophane and imogolite at Flakaliden indicates that this could
be the case, at least at Flakaliden.

Regarding pyroxene, XRPD might have failed to detect and quantify pyroxene due to their low abundancies at
Flakaliden (Casetou-Gustafson et al., 2018). Analytical limitations of XRPD would thus imply that $W_{XRPD}$ of  Ca
might be underestimated at Flakaliden and Asa. However, in the absence of XRPD detection it is also possible that
$M_{A2M-reg}$ overestimated the  pyroxene contents at  Flakaliden.  Thus, apatite and pyroxene added relatively large
uncertainties to the weathering estimates of Ca at Flakaliden due to the fact that they have a low abundance in
combination with very high dissolution rates. Furthermore, the overestimation of the slowly weatherable mineral
illite by $M_{A2M-reg}$ (Casetou-Gustafson et al., 2018) resulted in an underestimation of Ca release by $W_{A2M-reg}$ at
Flakaliden, since less Ca was allocated to the more weatherable minerals. Although, it should also be noted
parenthetically that Ca can only occur as an exchangeable cation in illite, it is not an element that occurs as part of
the illite structure, such that the 'illite' composition used in PROFILE may need some revision.

### 4.3.3 Magnesium release rates

At both study sites, a large number of Mg-containing minerals contributed to the release of Mg, but amphibole
was the predominant mineral according to $W_{XRPD}$ and $W_{A2M-site}$. The only significant discrepancy in Mg release
rates was revealed for $W_{A2M-site}$ at Asa, which resulted in an overestimation by 41 %. This overestimation was an
effect of underestimated  contribution from amphibole in combination with  overestimated contributions from
hydrobiotite and trioctahedral mica. This result for Asa supported our second hypothesis. At Flakaliden, $W_{A2M-site}$
produced the same shift in the contribution of Mg by minerals, but the net effect was a very small and non-
significant discrepancy to $W_{XRPD}$. As was noted for Ca, the different outcomes of using site-specific mineralogies
at Asa and Flakaliden has no systematic underlying pattern.
Using PROFILE based on regional mineralogy resulted in surprisingly low and non-significant discrepancies in
Mg release rate, despite both the qualitative and quantitative mineralogies being very different from XRPD,
particularly at Flakaliden. For example, both pyroxene and illite were included in $M_{A2M-reg}$, but not in $M_{XRPD}$. Thus,
at Flakaliden, the overestimation of illite in $M_{A2M-reg}$ caused an underestimation of Mg release rates comparable to
the underestimation of Ca release rates.





### 4.3.4 Potassium release rates

Weathering of K-feldspar was the most important source of K release by PROFILE regardless of the different types of mineralogy input. Casetou-Gustafson et al. (2018) showed a strong negative correlation between $M_{A2M-reg}$ and $M_{XRPD}$ for two of the major K-bearing minerals observed at both study sites, i.e., illite and K-feldspar. Contrary to our second hypothesis, the results of the present study demonstrate that over-or underestimation of $W_{A2M-reg}$ compared to $W_{XRPD}$ cannot be explained by significant negative correlation of illite and K-feldspar in $M_{A2M-reg}$. However, this is likely related to the fact that illite and K-feldspar have the lowest and also quite similar dissolution rates among minerals included in PROFILE (i.e. the highest dissolution coefficients, Table 3). Although very different inputs in relation to K bearing minerals produced very similar outputs, we note that this appears contradictory to differences in the behaviour of K-feldspars and K-micas as sources of K via weathering to plants as reviewed for example by Thompson and Ukrainczyk (2002).

### 5. Concluding remarks

- Based on comparing the full solution span of normative mineralogy from the A2M program to measured reference mineralogy from XRPD overall similar weathering rates were generated by the different mineralogical inputs to the PROFILE model. However, the underlying contributions from different minerals to the overall rates differed markedly. Although the similarity of overall rates lends some support to the use of normative mineralogy as input to weathering models, the details of the comparison reveal potential short-comings and room for improvements in the use of normative mineralogies.

- Compared with regional mineralogy, weathering rates based on site-specific mineralogy were more comparable to the reference rates generated from XRPD mineralogy, in line with hypothesis 1, at one of the study sites (Flakaliden), but not at the other (Asa). Thus, although intuitively the more detailed site specific quantitative mineralogy data might be expected to give more comparable results, this is not supported by this study.

- For Ca and Mg the differences between weathering rates based on different mineralogies could be explained by differences in the content of some specific Ca- and Mg-bearing minerals, e.g. amphibole, apatite, pyroxene and illite. Improving the precision in the content and presence versus absence of some of these minerals would reduce weathering uncertainties.

- High uncertainties in mineralogy, due for example to different A2M assumptions, had surprisingly low effect on the weathering from Na- and K-bearing minerals. This can be explained by the fact that the weathering rate constants for the minerals involved, e.g. the plagioclase feldspars and K-feldspar and dioctahedral micas, are similar in PROFILE such that they compensate each other in the overall weathering rate outputs for these elements, a situation that is unlikely to reflect reality.

- For more in-depth analysis of the uncertainties in weathering rates caused by mineralogy, the rate coefficients of minerals should be revisited and their uncertainties assessed. A revision of rate constants could lead to results more in line with hypothesis 1.

### 6. Authors contribution

Authors contributed to the study as in the following: S. Casetou-Gustafson: study design, data treatment, PROFILE model analyses, interpretation and writing. C. Akselsson: study design, PROFILE model development,



interpretation and writing. B.A. Olsson: study design, interpretation and writing. S. Hillier: interpretation and
writing.

**556 7. Acknowledgements**

Financial support from The Swedish research Council for Environment, Agricultural Sciences and Spatial Planning
(212-2011-1691) (FORMAS) (Strong Research Environment, QWARTS) and the Swedish Energy Agency
(p36151-1). Stephen Hillier acknowledges support of the Scottish Government's Rural and Environment Science
and Analytical Services Division (RESAS). We thank Salim Belyazid for his contribution to the design of the
study.

**562 8. References**

Aagaard, P., and Helgeson, H. C.: Thermodynamic and kinetic constraints on rection rates among minerals and
aqueous solutions:1.Theoretical considerations, Am. J. Sci., 282, 237-285, 10.2475/ajs.282.3.237, 1982.
Akselsson, C., Westling, O., Sverdrup, H., and Gundersen, P.: Nutrient and carbon budgets in forest soils as
decision support in sustainable forest management, Forest Ecol Manag, 238, 167-174, 2007a.
Akselsson, C., Westling, O., Sverdrup, H., Holmqvist, J., Thelin, G., Uggla, E., and Malm, G.: Impact of harvest
intensity on long-term base cation budgets in Swedish forest soils, Water, Air, & Soil Pollution: Focus, 7, 201-
210, 2007b.
Akselsson, C., Olsson, J., Belyazid, S., and Capell, R.: Can increased weathering rates due to future warming
compensate for base cation losses following whole-tree harvesting in spruce forests?, Biogeochemistry, 128, 89-

572 105, 2016.

Albaugh, T. J., Bergh, J., Lundmark, T., Nilsson, U., Stape, J. L., Allen, H. L., and Linder, S.: Do biological
expansion factors adequately estimate stand-scale aboveground component biomass for Norway spruce? Forest
Ecol Manag, 258, 2628-2637, 2009.
Allen, B. L., and Hajek, B. F.: Mineral occurrence in soil environments, in: Minerals in Soil Environments, edited
by: Dixon, J. B., and Weed, S. B., Soil Science Society of America Inc., Madison, no. 1, 199-278, 1989.
Andrist-Rangel, Y., Simonsson, M., Andersson, S., Öborn, I., and Hillier, S.: Mineralogical budgeting of
potassium in soil: a basis for understanding standard measures of reserve potassium, Journal of plant nutrition and
soil science, 169, 605-615, 2006.
Bergh, J., Linder, S., Lundmark, T., and Elfving, B.: The effect of water and nutrient availability on the
productivity of Norway spruce in northern and southern Sweden, Forest Ecol Manag, 119, 51-62, 10.1016/s0378-
1127(98)00509-x, 1999.
Blum, A. E., and Stillings, L. L.: Feldspar dissolution kinetics, in: Chemical Weathering Rates of Silicate Minerals,
edited by: White, A. F., and Brantley, S. L., Reviews in Mineralogy, Mineralogical Soc Amer, Chantilly, 291-351,

586 1995.



Casetou-Gustafson, S., Hillier, S., Akselsson, C., Simonsson, M., Stendahl, J., and Olsson, B. A.: Comparison of
measured (XRPD) and modeled (A2M) soil mineralogies: A study of some Swedish forest soils in the context of
weathering rate predictions, Geoderma, 310, 77-88, 2018.
Chou, L., and Wollast, R.: Steady-state kinetics and dissolution mechanisms of albite, Am. J. Sci., 285, 963-993,
10.2475/ajs.285.10.963, 1985.
de Jong, J., Akselsson, C., Egnell, G., Löfgren, S., and Olsson, B. A.: Realizing the energy potential of forest
biomass in Sweden–How much is environmentally sustainable?, Forest Ecol Manag, 383, 3-16, 2017.
Froberg, M., Grip, H., Tipping, E., Svensson, M., Stromgren, M., and Kleja, D. B.: Long-term effects of
experimental fertilization and soil warming on dissolved organic matter leaching from a spruce forest in Northern
Sweden, Geoderma, 200, 172-179, 10.1016/j.geoderma.2013.02.002, 2013.
Futter, M. N., Klaminder, J., Lucas, R. W., Laudon, H., and Kohler, S. J.: Uncertainty in silicate mineral
weathering rate estimates: source partitioning and policy implications, Environ Res Lett, 7, Artn 024025
599 10.1088/1748-9326/7/2/024025, 2012.

Hellsten, S., Helmisaari, H. S., Melin, Y., Skovsgaard, J. P., Kaakinen, S., Kukkola, M., Saarsalmi, A., Petersson,
H., and Akselsson, C.: Nutrient concentrations in stumps and coarse roots of Norway spruce, Scots pine and silver
birch in Sweden, Finland and Denmark, Forest Ecol Manag, 290, 40-48, 10.1016/j.foreco.2012.09.017, 2013.
Hillier, S.: Use of an air brush to spray dry samples for X-ray powder diffraction, Clay Miner, 34, 127-127, 1999.
Hillier, S.: Quantitative analysis of clay and other minerals in sandstones by X-ray powder diffraction (XRPD),
Clay mineral cements in sandstones, 34, 213-251, 2003.
Hodson, M. E., Langan, S. J., and Wilson, M. J.: A sensitivity analysis of the PROFILE model in relation to the
calculation of soil weathering rates, Appl Geochem, 11, 835-844, Doi 10.1016/S0883-2927(96)00048-0, 1996.
Jönsson, C., Warfvinge, P., and Sverdrup, H.: Uncertainty in predicting weathering rate and environmental stress
factors with the PROFILE model, Water, Air, and Soil Pollution, 81, 1-23, 1995.
Karlsson, P.-E., Ferm, M., Hultberg, H., Hellsten, S., Akselsson, C., and Karlsson, G. P.: Totaldeposition av kväve
till skog, IVL Swedish Environmental Research Institute, Stockholm, Sweden 37, 2012.
Karlsson, P.-E., Ferm, M., Hultberg, H., Hellsten, S., Akselsson, C., and Karlsson, G. P.: Totaldeposition av
baskatjoner till skog, IVL Swedish Environmental Research Institute, Stockholm, Sweden 65, 2013.
Kleeberg, R.: Results of the second Reynolds Cup contest in quantitative mineral analysis, International Union of
Crystallography. Commission on Powder Diffraction Newsletter, 30, 22-24, 2005.
Kolka, R. K., Grigal, D., and Nater, E.: Forest soil mineral weathering rates: use of multiple approaches,
Geoderma, 73, 1-21, 1996.



Koseva, I. S., Watmough, S. A., and Aherne, J.: Estimating base cation weathering rates in Canadian forest soils
using a simple texture-based model, Biogeochemistry, 101, 183-196, 10.1007/s10533-010-9506-6, 2010.
Kronnäs, V., Akselsson, C., and Belyazid, S.: Dynamic modelling of weathering rates – Is there any benefit over
steady-state modelling? SOIL, 2019.
Langan, S. J., Sverdrup, H. U., and Coull, M.: The calculation of base cation release from the chemical weathering
of Scottish soils using the PROFILE model, Water, air, and soil pollution, 85, 2497-2502, 1995.
Linder, S.: Foliar analysis for detecting and correcting nutrient imbalances in Norway spruce, Ecological Bulletins,
626  178-190, 1995.

Lofgren, S., Aastrup, M., Bringmark, L., Hultberg, H., Lewin-Pihlblad, L., Lundin, L., Karlsson, G. P., and
Thunholm, B.: Recovery of Soil Water, Groundwater, and Streamwater From Acidification at the Swedish
Integrated Monitoring Catchments, Ambio, 40, 836-856, 10.1007/s13280-011-0207-8, 2011.
McCarty, D. K.: Quantitative mineral analysis of clay-bearing mixtures: the "Reynolds Cup" contest, IUCr CPD
Newsletter, 27, 12-16, 2002.
Olsson, M., Rosén, K., and Melkerud, P.-A.: Regional modelling of base cation losses from Swedish forest soils
due to whole-tree harvesting, Appl Geochem, 8, 189-194, 1993.
Omotoso, O., McCarty, D. K., Hillier, S., and Kleeberg, R.: Some successful approaches to quantitative mineral
analysis as revealed by the 3rd Reynolds Cup contest, Clay Clay Miner, 54, 748-760, 2006.
Posch, M., and Kurz, D.: A2M - A program to compute all possible mineral modes from geochemical analyses,
Comput Geosci, 33, 563-572, 10.1016/j.cageo.2006.08.007, 2007.
R, C. T.: R: A Language and Environment for Statistical Computing. R Foundation for Statistical Computing,
Vienna, Austria, 2016.
Raven, M. D., and Self, P. G.: Outcomes of 12 years of the Reynolds Cup quantitative mineral analysis round
robin, Clay Clay Miner, 65, 122-134, 2017.
Röser, D., Asikainen, A., Raulund-Rasmussen, K., and Møller, I.: Sustainable use of wood for energy—a synthesis
with focus on the Nordic–Baltic region. Springer, Berlin, 2008.
Simonsson, M., and Berggren, D.: Aluminium solubility related to secondary solid phases in upper B horizons
with spodic characteristics, Eur J Soil Sci, 49, 317-326, 1998.
Staelens, J., Houle, D., De Schrijver, A., Neirynck, J., and Verheyen, K.: Calculating dry deposition and canopy
exchange with the canopy budget model: Review of assumptions and application to two deciduous forests, Water
Air Soil Poll, 191, 149-169, 10.1007/s11270-008-9614-2, 2008.
Starr, M., Lindroos, A.-J., Tarvainen, T., and Tanskanen, H.: Weathering rates in the Hietajärvi Integrated
Monitoring catchment, 1998.

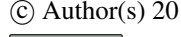



Stendahl, J., Lundin, L., and Nilsson, T.: The stone and boulder content of Swedish forest soils, Catena, 77, 285-
652    291, 2009.

Stendahl, J., Akselsson, C., Melkerud, P.-A., and Belyazid, S.: Pedon-scale silicate weathering: comparison of the
PROFILE model and the depletion method at 16 forest sites in Sweden, Geoderma, 211, 65-74, 2013.
Sverdrup, H., and Warfvinge, P.: Calculating Field Weathering Rates Using a Mechanistic Geochemical Model
Profile, Appl Geochem, 8, 273-283, Doi 10.1016/0883-2927(93)90042-F, 1993.
Sverdrup, H.: Geochemistry, the key to understanding environmental chemistry, Sci Total Environ, 183, 67-87,
658    1996.

Sverdrup, H., and Rosen, K.: Long-term base cation mass balances for Swedish forests and the concept of
sustainability, Forest Ecol Manag, 110, 221-236, 1998.
Thompson, M. L., and Ukrainczyk, L.: Micas, Soil mineralogy with environmental applications, edited by: Dixon,
J. B., and Schulze, D. G., Soil Science Society of America Inc., Madison, 431-466 pp., 2002.
Warfvinge, P., and Sverdrup, H.: Critical loads of acidity to Swedish forest soils: methods, data and results, Lund
University, 1995.
Weil, R. R., and Brady, N. C.: Soil phosphorus and potassium, in: The nature and properties of soils, Ed. 15,
Pearson Education, Upper Saddle River, USA, 2016.
Whitfield, C., Watmough, S., Aherne, J., and Dillon, P.: A comparison of weathering rates for acid-sensitive
catchments in Nova Scotia, Canada and their impact on critical load calculations, Geoderma, 136, 899-911, 2006.
Wikstrom, P., Edenius, L., Elfving, B., Eriksson, L. O., Lamas, T., Sonesson, J., Ohman, K., Wallerman, J., Waller,
C., and Klinteback, F.: The Heureka forestry decision support system: an overview, Mathematical and
Computational Forestry and Natural Resources Sciences, 3, 87-94, 2011.
Viro, P. J.: On the determination of stoniness, Communicationes Instituti Forestalis Fenniae, 40, 23, 1952.
Yu, L., Belyazid, S., Akselsson, C., van der Heijden, G., and Zanchi, G.: Storm disturbances in a swedish forest—
A case study comparing monitoring and modelling, Ecol Model, 320, 102-113, 2016.
Yu, L., Zanchi, G., Akselsson, C., Wallander, H., and Belyazid, S.: Modeling the forest phosphorus nutrition in a
southwestern Swedish forest site, Ecol Model, 369, 88-100, 2018.
Zanchi, G., Belyazid, S., Akselsson, C., and Yu, L.: Modelling the effects of management intensification on
multiple forest services: a Swedish case study, Ecol Model, 284, 48-59, 2014.
Zetterberg, T., Kohler, S. J., and Lofgren, S.: Sensitivity analyses of MAGIC modelled predictions of future
impacts of whole-tree harvest on soil calcium supply and stream acid neutralizing capacity, Sci Total Environ,
494, 187-201, 10.1016/j.scitotenv.2014.06.114, 2014.





**Table 1**. Characteristics of the study sites.

| Site | Asa | Flakaliden |
|---|---|---|
| Coordinates [a] | 57º 08' N; 14º 45´E | 64º 07'N; 19º 27'E |
| Elevation (m a.s.l.)[a] | 225-250 | 310-320 |
| Mean annual precipitation (mm)[b] | 688 | 523 |
| Mean annual air temperature (°C )[b] | 5.5 | 1.2 |
| Bedrock[c] | Acidic intrusive rock | Quartz-feldspar-rich sedimentary rock |
| Soil texture[d] | Sandy loam | Sandy loam |
| Type of quaternary deposit[d] | Sandy loamy till | Sandy loamy till |
| Soil moisture regime (Soil taxonomy)[e] | Udic | Udic |
| Soil type (USDA soil taxonomy)[e] | Spodosols | Spodosols |
| Region/province[f] | 3 | 1 |

[a] Bergh et al. 2005

[b] Long-term averages of annual precipitation and temperature data (1961-1990) from nearest SMHI meteorolgical stations (Asa: Berg; Flakaliden: Kulbäcksliden)

[c] SGU bedrock map (1:50000)

[d] Soil texture based on own  particle size distribution analysis by wet sieving according to ISO 11277

[e] USDA Soil Conservation service, 2014

[f] Warfvinge and Sverdrup (1995)






**Table 2.** PROFILE parameter description.

| Parameter | Description | Unit | Source |
|---|---|---|---|
| Temperature | Site | °C | Measurements from nearby SMHI stations |
| Precipitation | Site | m yr | Measurements from nearby SMHI stations |
| Total deposition | Site | $mmol_c$ $m^{-2}$ $yr^{-1}$ | Measurements of open field and throughfall deposition available from nearby Swedish ICP Integrated Monitoring Sites |
| BC net uptake | Site | $mmol_c$ $m^{-2}$ $yr^{-1}$ | Previously measured data from Asa and Flakaliden: Element concentration in biomass from Linder (unpublished data). Biomass data from Heureka simulations. |
| N net uptake | Site | $mmol_c$ $m^{-2}$ $yr^{-1}$ | Previously measured data from Asa and Flakaliden: Element concentration in biomass from Linder (unpublished data). Biomass data from Heureka simulations. |
| BC in litterfall | Site | $mmol_c$ $m^{-2}$ $yr^{-1}$ | Literature data for element concentrations from Hellsten et al. 2013 |
| N in litterfall | Site | $mmol_c$ $m^{-2}$ $yr^{-1}$ | Literature data for element concentrations from Hellsten et al. 2013 |
| Evapofraction | Site | Fraction | Own measurements and measurements from nearby Swedish Integrated Monitoring Sites |
| Mineral surface area | Soil | $m^2$ $m^{-3}$ | Own measurements used together with Eq. 5.13 in Warfvinge and Sverdrup (1995) |
| Soil bulk density | Soil | kg $m^{-3}$ | Own measurments |
| Soil moisture | Soil | $m^3$ $m^{-3}$ | Based on paragraph 5.9.5 in Warfvinge and Sverdrup (1995) |
| Mineral composition | Soil | Weight fraction | Own measurments |
| Dissolved organic carbon | Soil | mg $l^{-1}$ | Previously measured data from Asa and Flakaliden: Measurements for B-horizon from Harald Grip and previously measured data from Fröberg et al. 2013 |
| Aluminium solubility coefficient | Soil | kmol $m^{-3}$ | Own measurements for total organic carbon and oxalate extractable aluminium together with function developed from previously published data (Simonsson and Berggren, 1998) |
| Soil solution CO2 partial pressure | Soil | atm. | Base on paragraph 5.10.2 in Warfvinge and Sverdrup (1995) |







**Table 3** Mineral dissolution rate coefficients ($kmol_c\ m^{-2}\ s^{-1}$) used in PROFILE for the reactions with $H^+$, $H_2O$,
$CO_2$ and organic ligands ($R^-$) (Warfvinge and Sverdrup, 1995).

| Mineral | pkH | pkH2O | pkCO2 | pKR |
|---|---|---|---|---|
| Pyroxene | 12.3 | 17.5 | 15.8 | 14.4 |
| Apatite | 12.8 | 15.8 | 15.8 | 19.5 |
| Amphibole | 13.3 | 16.3 | 15.9 | 14.4 |
| Epidote | 14 | 17.2 | 16.2 | 14.4 |
| Plagioclase | 14.6 | 16.8 | 15.9 | 14.7 |
| K-Feldspar | 14.7 | 17.2 | 16.8 | 15 |
| Biotite | 14.8 | 16.7 | 15.8 | 14.8 |
| Chlorite | 14.8 | 17 | 16.2 | 15 |
| Vermiculite | 14.8 | 17.2 | 16.2 | 15.2 |
| Muscovite and Illite | 15.2 | 17.5 | 16.5 | 15.3 |




















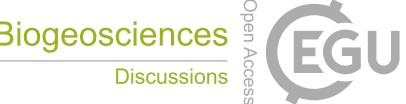

**Figure captions**
**Figure 1.** The first scenario for describing the effect of mineralogy on weathering rates in the upper mineral soil
for a specific soil profile (a) happens when the PROFILE weathering rate based on XRPD (reference weathering
rates) is not contained in the weathering range produced using PROFILE in combination with the full A2M
solution space. There are two possible explanations of why a significant discrepancy introduces an uncertainty
range, i.e. (1) due to uncertainties related to the mineralogical A2M input and (2) due to uncertainties related to
the limitation of the XRPD method itself (i.e. detection limit). The second scenario (b) occurs when the reference
weathering rate is contained in the full A2M weathering span. In this case we speak of 'non-significant
discrepancies'.
**Figure 2.** Comparison of PROFILE weathering rates of base cations ($mmol_c$ $m^{-2}$ $yr^{-1}$) at Asa (a) and Flakaliden
(b) sites in the 0–50 cm horizon based on XRPD mineralogy (vertical dashed lines) with PROFILE weathering
rates based on one thousand random regional A2M mineralogies versus one thousand random site-specific A2M
mineralogies. Data presented are from four different soil profiles per site. Regional graph for soil profile 10B at
Flakaliden is missing since A2M did not calculate 1000 solutions for soil layer 20-30, due to "Non-positive
solution".
**Figure 3.** Root-mean square error (RMSE) of average PROFILE weathering rates ($mmol_c$ $m^{-2}$ $yr^{-1}$) of one
thousand A2M mineralogies per soil layer, compared to weathering rates based on XRPD mineralogy per soil
layer. Comparisons are based on the total weathering per element (A) and on the sum of mineral contributions to
total weathering per element (B). RMSE describes the prediction accuracy for a single soil layer.
**Figure 4.** Comparison of PROFILE weathering rates based on XRPD mineralogy ($mmol_c$ $m^{-2}$ $yr^{-1}$) with
PROFILE weathering rates based on regional A2M mineralogy (upper figures) versus site-specific mineralogy
(lower figures). Each data point represents a mean of one thousand PROFILE weathering rates for a specific soil
depth of one of 4 soil profiles per site.
**Figure 5.** Comparison of sums of PROFILE base cation weathering rates for different minerals in the upper
mineral soil (0-50 cm) based on XRPD mineralogy and the average PROFILE base cation weathering rate (i.e.
based on one thousands input A2M mineralogies per mineral) according to the two normative mineralogical
methods and for each study site (i.e. Asa site-sepcific, Flakaliden site-specific, Asa regional, Flakaliden
regional). For $W_{A2M}$, relative error (% of $W_{XRPD}$ estimate) are given at the end of each bar to illustrate the
average deviation of $W_{A2M}$ and $W_{XRPD}$ in the upper mineral soil. *=significant discrepancy as defined in section
2.7. Vrm1=Trioctahedral vermiculite; Vrm2=Dioctahdreal vermiculite. Information on chemical compositions of
minerals are given in Table S5 and S6.
**Figure 6.** Sum of weathering rates ($mmol_c$ $m^{-2}$ $year^{-1}$) in the upper mineral soil (0–50 cm) for Ca, Mg, K and Na
for Asa (A) and Flakaliden (B) and for nearby sites from Stendahl et al. (2013) (Bodafors, Lammhult,
Svartberget and Vindeln).
















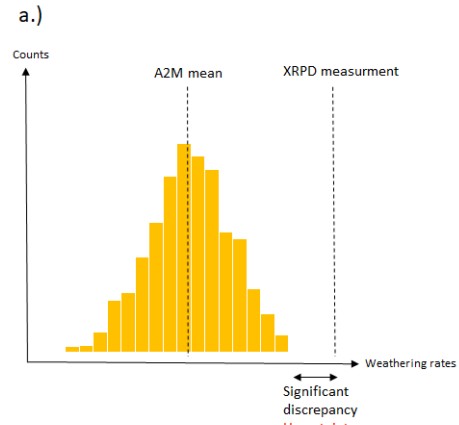 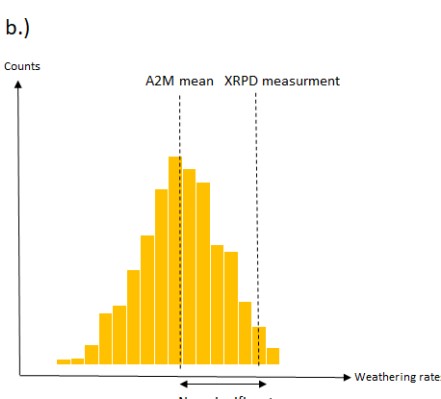


Figure 1a,b









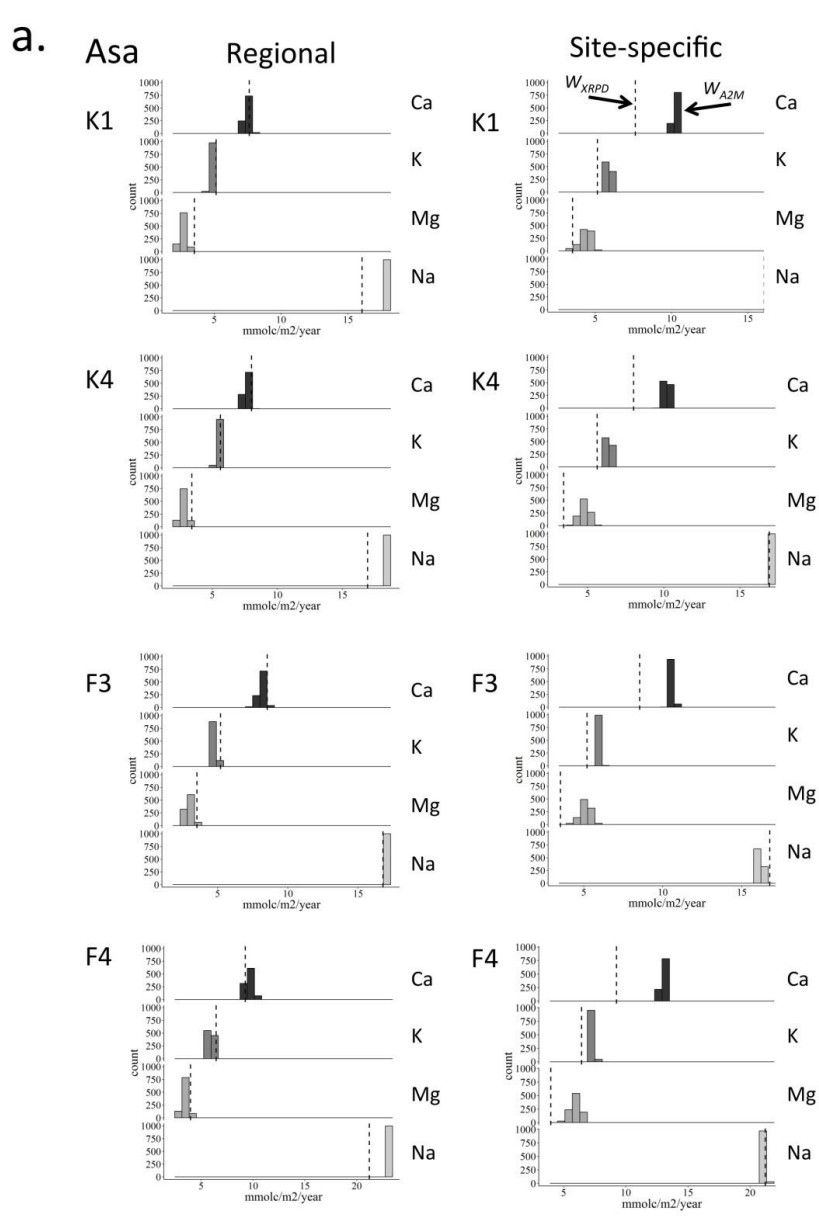


763          Figure 2a



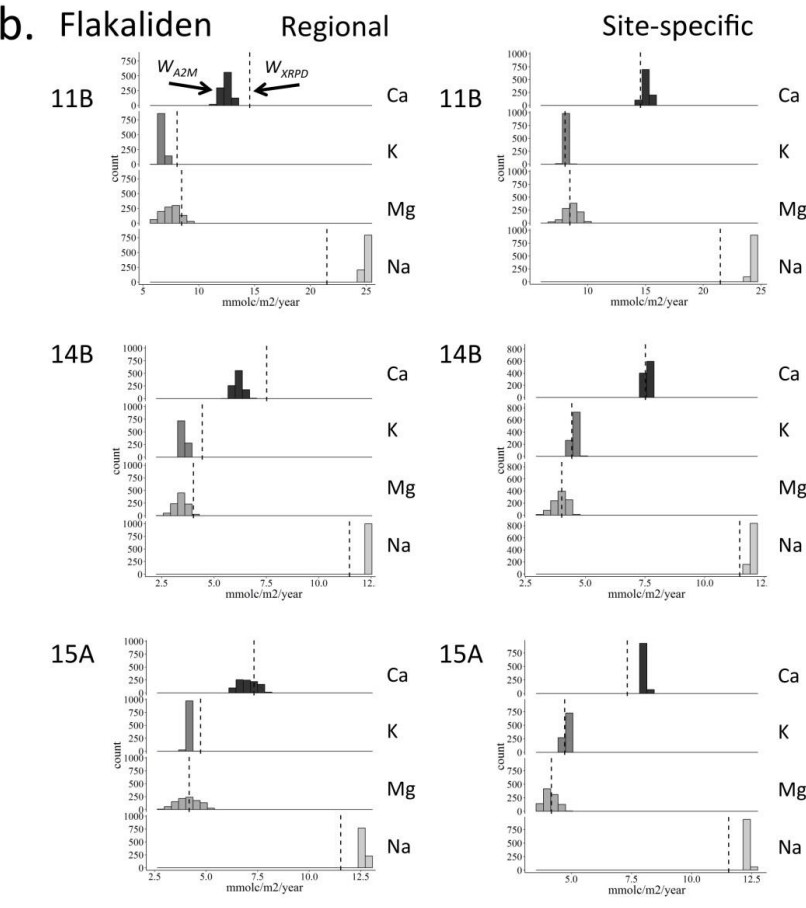

765        Figure 2b





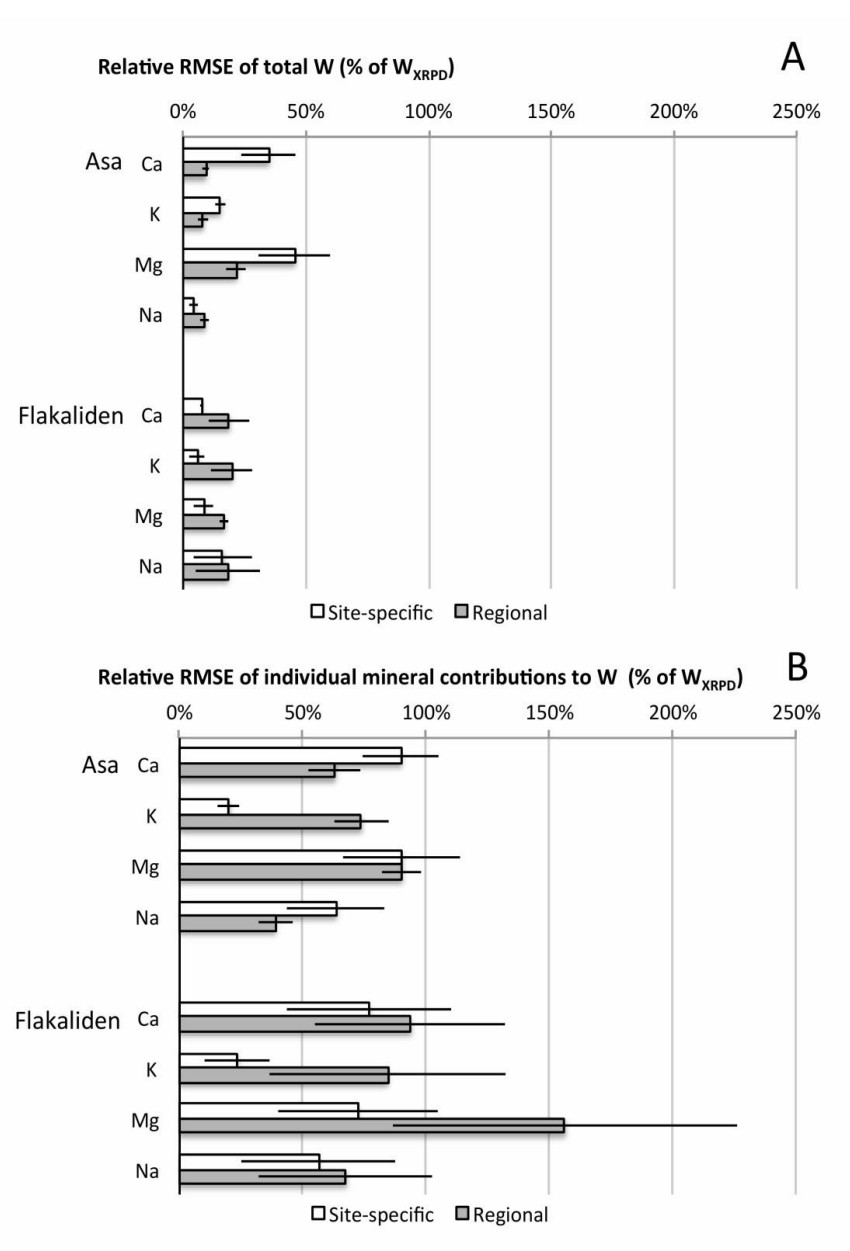


767  Figure 3, A, B







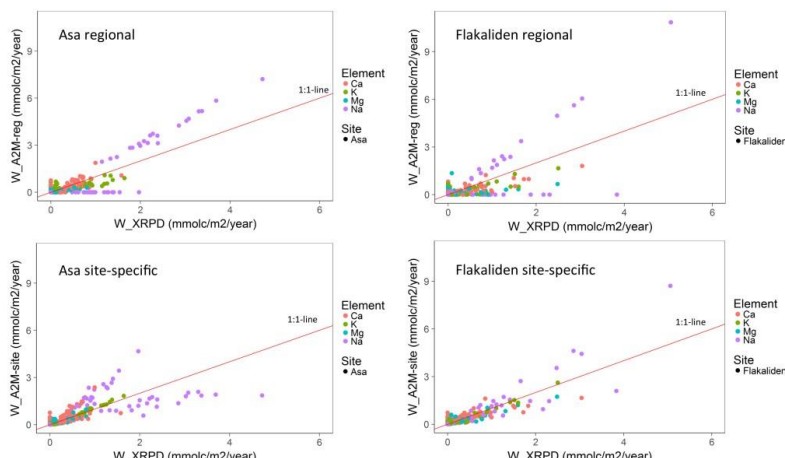

Figure 4






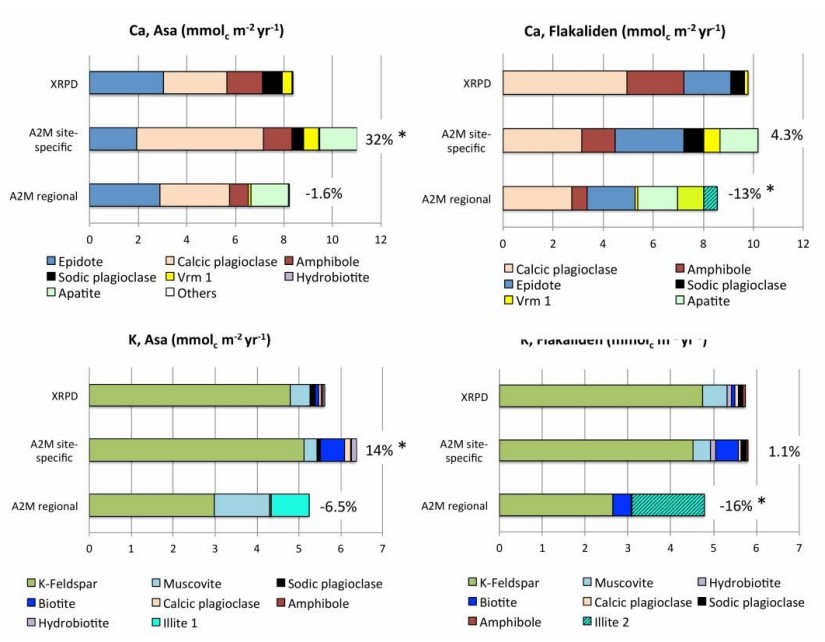


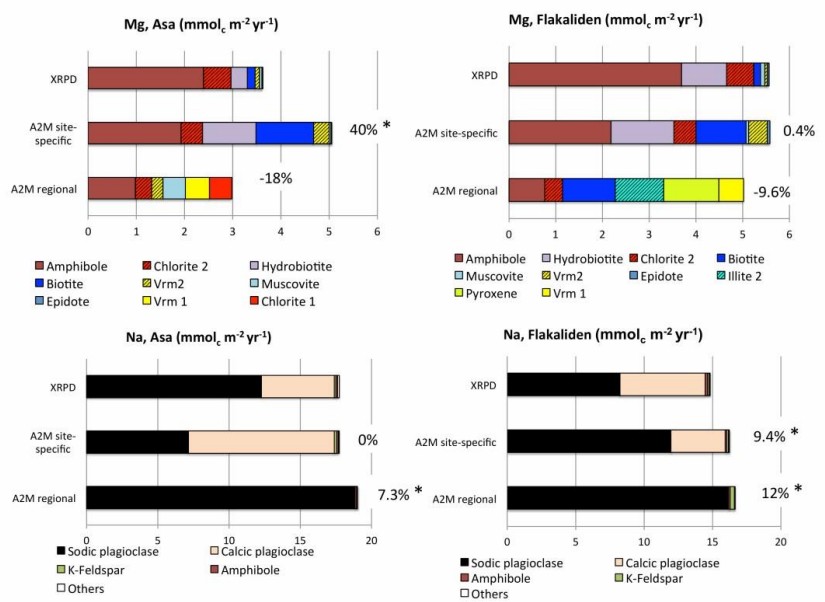


775        Figure 5





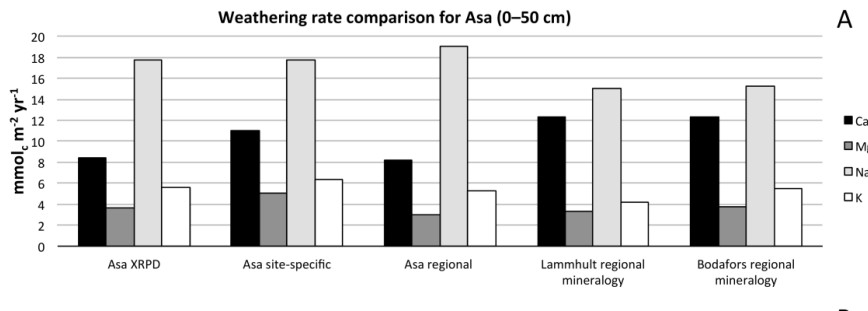

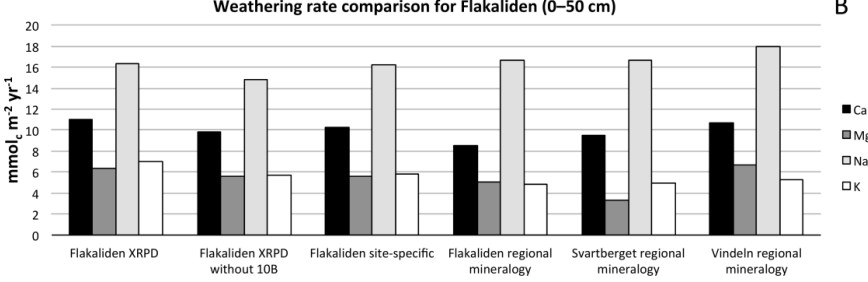


779        Figure 6 a, b