# Peer review of "The importance of mineral determinations to PROFILE base"

_Biogeosciences, 2018_

## Referee Comment (RC1) · Anonymous Referee #1 · 26 Jan 2019

Abstract I found the abstract quite hard to understand as I am not familiar with A2M – I suspect that other readers may have a similar issue.

Introduction Whilst the authors do highlight the sensitivity of the PROFILE model to the mineralogy inputs as determined by Hodson et al (1996) and Jonsson et al, (1995) any discussion of the PROFILE model and its use probably merits reference to the paper of Hodson et al (1997) Water, Air and Soil Pollution paper published by workers at what is now the James Hutton Institute (i.e. the same institute where the current authors work) which highlights significant problems with the model including (relevant to the paper reviewed here) issues related to assumed mineral composition within the default

[Figure]

PROFILE database

Hypothesis (1) – I struggled with this – the "reference weathering rates are those determined using measured mineralogies. In hypothesis 1 it is stated that rates calculated using site specific mineralogies are more similar to the reference rates than those using regional mineralogies but aren't the site specific mineralogies those actually measured, i.e. they are quantitative so the hypothesis makes no sense. I'm sure I must have simply misunderstood what the authors mean but I wonder wether they could try and clarify this.

Materials and methods Whilst values are kept constant between model runs to allow comparison the authors use Warfvinge and Sverdrups equation to calculate surface area. It is probably relevant to cite the work of Hodson et al (1998) Geoderma who showed that this equation was flawed in case others less familiar with the literature start to use this equation routinely. It also seems that rather than using the relative surface area of different minerals as required by the model the authors have used the relative weight % of the minerals. Almost everyone does this because the relative surface area of the different minerals present is almost impossible to determine.and for the comparison exercise this isn't an issue but again I think it should be acknowledged that in principle the "wrong" inputs have been used

Materials and Methods / Results / Discussion I'm a firm believer that all the data necessary to understand a paper should be in a paper, not in supplementary material or other papers. I may have missed it but I struggled to find a note of the regional mineralogy / bedrock type assumed for the sites and at list of the minerals determined in the actual mineralogical measurements. I think this should be added. I mention this because Art White (e.g. White et al, 2017, Geochim, Cosmochim Acta) did a lot of work looking at the importance of calcite in release of Ca during the weathering of granite, despite the presence of Ca-bearing plagioclases. In this paper there is discussion about the relative importance of plagioclase, apatite and epidote for Ca release in the profile model. Given the relative abundances and known dissolution rates of calcite,

apatite and epidote and White's findings I find it extremely unlikely that calcite wasn't important. Was it that no calcite was detected by the XRD / estimates of mineralogy? Is that likely given the rock types? I'd like to see a fuller discussion of the issue of trace concentrations of calcite resulting in high Ca release rates.

Concluding remarks Final bullet point seems to agree with the Hodson et al (1997) paper so again reference to that paper might be appropriate

---

## Referee Comment (RC2) · Schroeder (Referee) · 28 Jan 2019

bg-2018-522
Review comments by Paul A. Schroeder

Scientific significance:  Excellent (1)
Scientific quality: Excellent (1)
Presentation quality: Good (2)

1. Does the paper address relevant scientific questions within the scope of BG?
   - Yes
2. Does the paper present novel concepts, ideas, tools, or data?
   - yes
3. Are substantial conclusions reached?
   - Yes, but I would have broader impact if the results of weathering rate estimates were compared with estimates from other climate and lithologic regimes.
4. Are the scientific methods and assumptions valid and clearly outlined?
   - Yes, but the paper is reliant on the reader being familiar with XRPD and A2M quantitative methods and the modeling concepts of PROFILE.  The authors do a good job justifying their assumptions, but maybe a little more explanation would help the reader make the leap of faith to believe in their model.  Some might argue that using mean annual temperature and rainfall data in a kinetic model may not reflect the conditions for when the kinetics are the fastest (i.e., what time of year are the reactions taking place?).
5. Are the results sufficient to support the interpretations and conclusions?
   - Yes.
6. Is the description of experiments and calculations sufficiently complete and precise to allow their reproduction by fellow scientists (traceability of results)?
   - See response to 4 above.
7. Do the authors give proper credit to related work and clearly indicate their own new/original contribution?
   - Yes. However, if they are including comparisons to other climatic and lithologic regimes, then clearly, they would give attribution.
8. Does the title clearly reflect the contents of the paper?
   - Yes. However sorry, it's longer but maybe… "The importance of mineral determinations to base cation weathering release rates in Sweden: A case study using quantitative X-ray diffraction and elemental data and the kinetic model PROFILE."
9. Does the abstract provide a concise and complete summary?
   - Yes
10. Is the overall presentation well-structured and clear?
    - Yes
11. Is the language fluent and precise?
    - It's OK.  It would improve considerably with the aid of a good copy editor.

12. Are mathematical formulae, symbols, abbreviations, and units correctly defined and used?
    - Yes
13. Should any parts of the paper (text, formulae, figures, tables) be clarified, reduced, combined, or eliminated?
    - Fonts on the figure 2 axis labels are a bit small.
14. Are the number and quality of references appropriate?
    - Yes. However, see comment 4
15. Is the amount and quality of supplementary material appropriate?
    - The link to supplementary material did not work….

---

## Editor Comment (EC1) · Suzanne Anderson (Editor) · 21 Feb 2019

Summary: The abstract is dense, and therefore hard to follow. Hypothesis 1 lacks clarity, owing to incomplete descriptions of the differences between mineralogy inputs into the PROFILE model. Trace highly reactive phases (notably calcite) should perhaps be considered in this analysis.

Details: The topic of this manuscript is fairly straightforward, and yet it is extremely easy to get lost in the details while reading it. The central question is about errors that arise in model-based weathering rate calculations using an assumed regional mineralogy versus a site-specific mineralogy. The authors define a reference "truth" as

the model-based (using PROFILE) weathering rates calculated from measured quantitative mineralogy (with XRPD) and measured elemental compositions from the field sites. These reference rates are designated $W_{XRPD}$. They compare these reference "truth" rates with rates determined with PROFILE, but using two different modeled mineralogies. The first modeled mineralogy is apparently based on the same data as the "truth", but rather than directly using the XRPD and elemental composition data, they run A2M to produce 1000 different mineralogies consistent with the inputs. This is the site-specific case ($W_{A2M-site}$). The second modeled mineralogy is also derived using A2M, but from regional mineralogy descriptions (and the derived rates are designated $W_{A2M-reg}$).

The similarity of the reference "truth" rates ($W_{XRPD}$) the site-specific rates ($W_{A2M-site}$) is confusing, and could use more clarification. I see this as the most significant matter the authors should address.

I agree with comments of R1 that consideration of possible unmeasured trace phases, such as carbonates, should be made.

The PROFILE model is described as "the most used mechanistic tool to calculate steady state chemical weathering at the interface of soil minerals and their surrounding liquid solution". I do not think this claim is necessary or useful. It would be better to briefly explain how PROFILE works, and its long history, particularly in Sweden.

Figure 5 would be much easier to read if the order of color blocks in each bar was consistent, at least in right and left panels of the plot, if not across all panels of the plot. For one thing, then only one key would be required per pair of panels.

[Figure]

---

## Author Comment (AC1) · 4 Mar 2019

We would like to thank the referee for their helpful comments. Abstract: We will try to simplify the abstract. Introduction: We agree with the referee that it was an oversight not to refer to the work by Hodson et al. (1997) and we will cite it in the introduction. We also thank the referee for pointing out the difficulty of understanding hypothesis 1. We would like to clarify that site-specific mineralogy is determined in terms of the minerals identified and the chemical compositions of these minerals but not determined (directly) in terms of the abundance of these minerals, which is instead calculated by A2M. Although the definitions of 'site specific', 'regional' and 'measured' mineralogy are

given in the text of the introduction we propose also to add them to the definition section to aid clarity. Material and Methods: We agree with the referee and will cite Hodson et al. (1998) in the Material and Methods section. Since we did not mention Table S1a, S1b and S6 in section 2.5.1 and Table S5 in section 2.5.2, we should adjust for this and add a reference to those tables. Materials and Methods / Results / Discussion: We did not detect any calcite in our mineralogical analyses of the soil samples by XRD (line 224). The study by White has highlighted the importance of small amounts of calcite in intact granitoid rocks and its significance for Ca found in watershed studies. The mean value from the above-mentioned study by White is 0.25 wt. %, and the median 0.075 wt. % calcite. White also noted that in laboratory leaching experiments on the rocks they studied reactive calcite became exhausted after just 1.5 yr. Given the trace concentrations involved and the high solubility of calcite we doubt very much that calcite is or has been of any long-lived significance in the soil profiles studied, even though they are derived largely from rocks of granitic composition. Though we agree that Whites results indicate that calcite in the in-situ granitoid bedrock underlying the soils probably will contribute to Ca export from the catchment. In response, we propose to add discussion of the possible presence of traces of calcite to the discussion at the same point that we already discuss the possible presence of traces of apatite and pyroxene (lines 482-496). Conclusion: With regard to the suggestion of referencing Hodson et al. (1997) in the concluding remarks, we will try to include their criticism about the reaction rate coefficients.

―――――――――――――――――――――――――――

---

## Author Comment (AC2) · 4 Mar 2019

We would like to thank the referee for his helpful comments. With regard to point 3 and the suggestion of comparing our weathering rate estimates with estimates from other climate and lithologic regimes, we argue that our study was necessarily focused without scope to extend to other regimes. Since the focus of this manuscript is the soil mineralogical input to PROFILE, we do not believe a paragraph about the limitations of PROFILE with regard to other PROFILE input parameters would be adequate as suggested by the referee under point 4. However, we could add some lines and/or rephrase sentences in paragraph 2.6.1 and define the steady state concept. This is a

well-known and often used concept. Of course it has draw-backs compared to dynamic models, like ForSAFE, but it also has advantages (e.g. simpler, thus more transparent and easy to use). With regard to point 7 and as mentioned in our reply under point 4, we are not able to extend the study to climatic/lithological regimes. We thank the referee for his/her suggestion under point 8, however, but we prefer our shorter version. We agree with the referee about suggestions made under point 13, they can be enlarged.

---

## Author Comment (AC3) · 4 Mar 2019

We want to thank the editor Suzanne Andersson for her helpful comments. We agree with the editor and understand that the abstract needs to be improved. We also agree that hypothesis 1 can be misunderstood. We will clarify that site-specific mineralogy is determined in terms of the minerals identified and the chemical compositions of these minerals but not determined (directly) in terms of the abundance of these minerals, which is instead calculated by A2M. With regard to calcite, as per our response to the comment of Anonymous Reveiwer #1, we agree that it requires a brief mention and suggest that this could be done in paragraph 4.3.2. With regard to the details of the

manuscript, we would like to clarify that the first modeled mineralogy is based on only some of the same data, the minerals present and their chemical compositions, not their abundances. We would also like to clarify that our aim was to test whether or not PROI-FLE based on normative calculations performed with A2M, produce an output in closer agreement with PROFILE based on mineralogy obtained from XRPD, if site specific mineralogical data are used as input to A2M and PROFILE rather than regional data. The latter means that both mineral identity and mineral stoichiometry have been based on measured data and given as an input to A2M. A2M then calculated the quantitative mineralogy based on this site-specific input. The reference quantitative mineralogy was solely determined by analytical measurements. Furthermore, we agree with the suggestion of briefly explaining how PROFILE works, and its long history, particularly in Sweden. We will make the required changes in the material and method section. With regard to figure 5, we agree that the layout could be improved. Minerals in the bars are now ranked from high-to-low abundance in the XRPD-bars. We are not sure the suggested change is the best for easy reading, but will remake alternative layouts, test them for readability and pick the best!

---

## Author Response (AR1)

[revised manuscript text omitted]

Figure 1a,b

[Figure]

Figure 2a

[Figure]

Figure 2b

[Figure]

[Figure]

Figure 3, A, B

[Figure]

Figure 4

[Figure]

[Figure]

Figure 5

**Resubmission letter**

Dear Prof. Andersson,

Thank you for providing the reviews and for your editorial comments on our manuscript and the opportunity to submit a revised version. Your suggestions (marked in grey color) were:

1) Clarification and simplification of the abstract
2) Clarify hypothesis 1
3) Discussion of the possible role of calcite or other reactive trace phases on weathering fluxes.
4) With regard to point 4 in RC 2, please describe the relationship between mean annual precipitation and temperature (used in the model) and the range in these values (which may actually be the more relevant drivers of weathering)
5) Briefly explain how PROFILE works
6) Improve presentation of Figure 5
Please explain your decisions with regard to all major comments of the reviewers.

Non-public comments to the Author:
There are other sources than the White paper that consider the role of highly reactive phases on weathering fluxes. Two that spring to mind are: Anderson et al., 2000, GCA (my own work, sorry--not intending to just congratulate myself!), and Jacobson et al., 2002, GCA (Vol. 66, No. 1, pp. 13–27, 2002)

We have carefully read through all the comments and have critically discussed possible changes. We hope that you will find all major points sufficiently addressed and that the manuscript is acceptable to you in revised form.

In terms of changes we have focused mainly on your suggestions:

Details of the changes are as follows in the manuscript with track and changes:

1. Abstract (Lines 18-43): We have reduced the length of the abstract by removing text about the detailed use of A2M. We also introduced new sentences in the beginning of the abstract, which should give the background of the study. Further, we have tried to clarify hypothesis one in the abstract.

2. Introduction (Lines 96-99): We have included a better short description of the PROFILE model in line with point 5 of your suggestions.

3. Introduction (Lines 166-169): We have attempted to clarify hypothesis one in line with point 2 of
your suggestions.

4. Material and Methods (Lines 280-285): In line with point 4 of your suggestions, we have included a
point about the use of average annual temperature and precipitation data in PROFILE.

5. Discussion (Lines 517, 530-537, 584): We have introduced a discussion about the possible role of
calcite on weathering fluxes and added White et al. (1996) and White et al. (2017) to the reference list
(Lines 730-734).

6. Figure 5: The presentation is improved.

Additional changes:

We have by mistake not been consistent in abbreviating journal names and in the handling of DOIs and
have corrected for this in the reference list where required.

We also remade equation 1 and 2, because there was a bracket too much and an *i* lacking (Line 317 and
331).

Concerning our decisions with respect to all major comments of the reviewers:

-Anonymous **referee 1:**

1: We have now cited the Hodson et al. (1997) publication in the introduction (Lines 109-110) and
included it in the reference list (Lines 664-665): With regard to mineralogy, Hodson et al. (1997)
referred mainly to errors in input data, and input data varies between different model runs, depending
on the user.

2. In line with the editor suggestion, we have tried to clarify hypothesis 1 (Lines 166-169).

3. We have included Table S1a, S1b and S6 in section 2.5.1 and Table S5 in section 2.5.2 (Lines 212,
217, 233, 231).

4. We have included the Hodson et al. (1998) publication in the material and method section (Line 261)
and hence in the reference list (Lines 660-661)).

5. In line with the editor suggestion, we have introduced a discussion about calcite (Lines 517, 530-537,
584).

6. With regard to the suggestion of referencing Hodson et al. (1997) in the concluding remarks, we have included
their criticism about the reaction rate coefficients in the last sentences of the discussion (Lines 568-569) and
added a reference to the reference list (Lines 713-714).

-**Referee 2** (Paul Schröder):

1: In line with the editor suggestion in point 4, we have given a better explanation of the use of annual
temperature and precipitation data in PROFILE (Lines 280-285).

In addition to the revised manuscript we have also provided a version with tracked changes, so that
the revisions are clearly visible. We have also been carefully through the whole text and made some
additional minor revisions which we believe aid clarity. We hope you will find our revised manuscript
acceptable for publication in Geoderma and look forward to hearing from you in due course.

On behalf of all authors, yours sincerely,

Sophie Casetou-Gustafson,

Corresponding author